# Equivariant Diffusion for The Inverse Radar Problem

## Abstract

Reconstructing 3D geometries from their radar signal is a complex inverse problem, often involving unique domain expertise and manual steps. Although deep learning approaches have emerged to address the automation challenges of this problem, there are still significant performance gaps due to non-unique solutions and partial observability. In this work, we explore the role of equivariant modeling in helping reduce uncertainty over potential 3D shape distributions measured via partial radar signals. We present a radar-conditioned equivariant latent diffusion model that uses a two-stage training approach. In the first stage, we learn equivariant latent representations of 3D shapes by training a SO(3)-equivariant encoder-decoder model using vector neuron architectures. During the second stage, we train an SO(3)-equivariant denoising diffusion model that operates over the learned latent geometry representations. We introduce an equivariant FiLM layer that enables conditioning of our diffusion model in Irreps space and thus ensures rotational equivariance throughout the generation process. Finally, we ensure equivariant latent representations of the conditioning radar signal by using a spherical CNN model. We show that our model predicts plausible 3D geometries consistent with the observed radar signatures. In addition, we demonstrate improved performance over other competitive non-equivariant baseline methods with respect to one of the reconstruction quality metrics and a sample diversity metric under full observability settings.

## 1 Introduction

Although radar has been used in several applications to sense the 3D world around us, the problem of reconstructing the 3D geometries captured by radar signals is largely underexplored. Often, radar signals are used to perform other inverse learning tasks, such as object classification, tracking, and task-specific feature extraction. In real applications, it is also common to have only partial radar observations that do not cover the full geometry of the objects, introducing significant uncertainty in the geometry reconstruction task.

Previous deep learning approaches have attempted to solve this problem (Muthukrishnan et al., 2023; Lundén & Koivunen, 2016; Wan et al., 2020), but significant performance gaps still persist. Due to the lack of large datasets and the high computational complexity required for this task, these traditional deep learning methods fall short. We address this challenge by employing geometric deep learning, which utilizes equivariant neural networks (Cohen et al., 2018). Equivariant neural networks use symmetries present in the data to learn more robust features and have been shown to require less data and training time compared to non-equivariant models. (Kohler et al., 2025) demonstrates the superiority of equivariant models for forward radar problems, where the goal is to predict radar signals from 3D geometry.

Due to the presence of aleatoric uncertainty in radar signals, it is difficult to solve this problem with deterministic models. Diffusion models seem to be a suitable choice to capture this uncertainty, but most existing state-of-the-art diffusion models for shape generation are conditioned on single-view images or partial point clouds rather than spherical radar signals (Chou et al., 2023; Zeng et al., 2022). Furthermore, they lack architectural components for conditioning using equivariant embeddings.

To address these challenges, we introduce a radar-conditioned equivariant latent diffusion model that can learn a probabilistic mapping of radar signatures to the corresponding 3D geometry. We make the following contributions:

- We introduce an equivariant FiLM layer for conditioning diffusion models on radar embeddings.

- We develop an SO(3) equivariant latent denoising diffusion model for 3D shape reconstruction from radar signals.

- We also show that our model outperforms other competitive baselines for F-score(1%) and TMD metrics.

## 2   RELATED WORK

**Equivariant Neural Networks.**   Equivariant neural networks are mathematically biased to respect the symmetry present in the input data(Cohen & Welling, 2016; Thomas et al., 2018; Fuchs et al., 2020). Due to the presence of symmetries in many tasks, they have been used in several domains to improve model performance and efficiency. (Esteves et al., 2018)(Cohen et al., 2018) uses it for 3D shape classification. (Liao & Smidt, 2022)(Liao et al., 2023) leverages the symmetries present in 3D atomic structures to predict the chemical properties of molecules. (Deng et al., 2021) developed a framework to build SO(3) equivariant networks to process point clouds. (Lei et al., 2023) Yang et al. (2024) uses the same framework to learn equivariant latent embedding from point clouds, which are later used for 3D object segmentation and diffusion policy for robotics. (Kohler et al., 2023) uses it to map 3D geometries to spherical signals and shows their successful application in predicting radar signals from 3D geometries. All of these models have shown better performance compared to other non-equivariant baselines. While several methods have used equivariant neural networks to process and extract features from 3D geometries, none of them have been used to generate 3D shapes from radar signals.

**3D Shape Reconstruction**   3D shape reconstruction has been an open computer vision problem for several years. Several methods have been developed to reconstruct 3D shapes from 2D images and partial point clouds. Recent approaches to 3D shape reconstruction from 2D images or partial inputs have largely focused on latent generative models and implicit representations. (Zeng et al., 2022) is based on a hierarchical variational autoencoder (VAE) framework with two denoising diffusion models for 3D shape generation. (Tang et al., 2024) uses an Auto-regressive auto-encoder to generate variable-length meshes from fixed-length latent codes. They also use the same latent codes to train a diffusion model for generating 3D meshes from 2D images or partial point clouds  (Chou et al., 2023). Many of these works use Deep Signed Distance Functions (SDFs)  (Park et al., 2019a) as implicit representations, where query points $\in \mathbb{R}^3$ are mapped to the shortest distance to a target surface, and the isosurface of the shape is the zero level-set of this collection of points. The main difference between the methods mentioned above and ours is that we use an equivariant neural network to denoise the latent codes for shape generation.

**Radar Modeling.**   Previous works in deep learning attempt to encode radar responses to characterize the target object. Some works encode the spatial information of the radar response using 1-dimensional convolutional neural networks (CNNs)  (Lundén & Koivunen, 2016; Wan et al., 2020), recurrent neural networks (RNNs) (Xu et al., 2019), or attention mechanisms  (Pan et al., 2022; Wan et al., 2020; Xu et al., 2019). These works typically focus on target object classification rather than full reconstructions. (Muthukrishnan et al., 2023) tackles reconstruction by designing a transformer-based architecture to encode the spatial and temporal aspects of the radar response and predict the full shape profile of roll-symmetric objects. While this strong shape prior improves performance, it restricts the model's ability to reconstruct objects with arbitrary shapes. This remains a more difficult task, particularly when the radar response is noisy and partially observable. Our method takes a different approach by using a spherical CNN (Cohen et al., 2018) to extract features from radar signals.

## 3   BACKGROUND

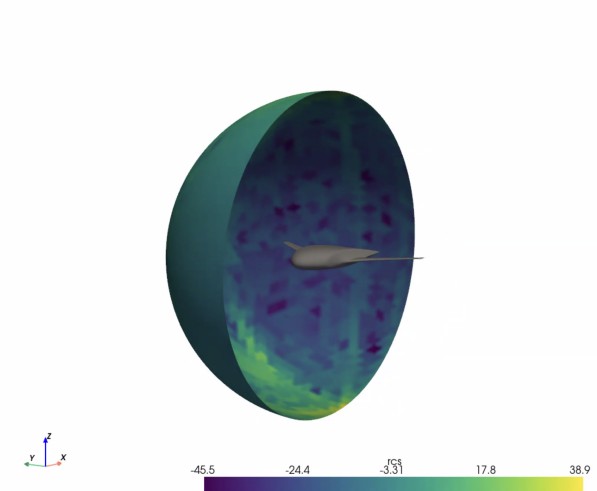

Figure 1: Illustration of the spherical radar signal of an airplane 3D shape for sensor viewing angles $[0, 180]$. Lighter, yellow colors represent higher amplitudes of the radar response.

**Problem setting**  We consider the problem of generating 3D objects from radar responses. We focus in particular on high-frequency radar waveforms, where the radar wavelength is much smaller than the object size. This type of radar is used in most commercial and defense-related applications. To calculate a radar response, we utilize Physical Optics (PO) (Balanis, 2012) to estimate how the waves scatter off a triangle of a mesh. Using the Geometric Theory of Diffraction (Keller, 1962), the entire mesh's scattering response from a viewing angle $u$ can be reduced to a sum of individual triangle scattering responses. We use *range profile*, a common view for radar, as input to our models. This is calculated by taking the Discrete Fourier Transform of the mesh's scattering response across each frequency.

Figure 1 demonstrates the spherical nature of a radar signal (as a range profile), where the sphere is parametrized by the viewing angle $u$ of a 3D geometry. In this work, we learn to reconstruct Signed Distance Functions (SDFs) (Park et al., 2019b) from such a spherical signal. SDFs are implicit representations that, for a given query point $X \in \mathbb{R}^3$, give us the distance of that query point from the surface of the shape. From a learned continuous SDF, we can reconstruct a shape by sampling several points in a bounding box and filtering out points that are close to the surface. We set the distance threshold to $0.01$. Any query point with an SDF value less than the threshold is considered to be on the surface of the shape.

**Equivariance**  Given a spherical radar signal, $r \in \mathbb{S}^2$, an inverse radar function $f : \mathbb{S}^2 \to \mathbb{R}^3$ will give a shape $P \in \mathbb{R}^3$ as the output, i.e., $f(r) = P$. If $r$ is rotated by $R \in SO(3)$, the output shape $P$ is also rotated by $R$, that is, $f(Rr) = RP$. This property of the inverse radar function is called equivariance. More precisely, the inverse radar function is equivariant with respect to all elements of the SO(3) group, including all possible rotations around the origin of the Euclidean space $\mathbb{R}^3$.

**Equivariant Neural Networks**  We use an SO(3) equivariant neural network in this work to respect the transformations in radar signals and output geometries mentioned above. We use two different equivariant neural network architectures.

For our SDF model, we use the Vector Neurons Deng et al. (2021) framework to generate SO(3) equivariant latent codes for 3D point clouds. Vector Neurons extends the traditional 1D scalars used in neural networks to 3D vectors. They also design a vector equivalent of all the traditional neural network layers, which are also equivariant to SO(3).

For our diffusion model, we use the e3nn library Geiger et al. (2022) to design our SO(3) equivariant denoising network. e3nn projects spherical signals to irreducible representations (irreps) of SO(3).

Once in the irreps space, we can use SO(3) preserving operations provided by e3nn to maintain equivariance throughout the network.

**Conditional Latent Diffusion Model** The denoising diffusion probabilistic model (DDPM) presented in Ho et al. (2020) uses a forward and reverse diffusion process to generate new data samples from an input distribution $q(x)$. During the forward process, we take a sampled data point $x_0$ and iteratively add Gaussian noise to it for $T$ time steps, resulting in $x_T$. During the reverse diffusion process, we do the opposite and iteratively denoise $x_T$ for $T$ time steps to obtain $X_0$. The reverse diffusion process is performed using a neural network model $p_\theta$. The training objective of DDPMs is to minimize the variational lower bound of the negative log-likelihood of the generated data that matches the true data distribution, over all timesteps 1 to T. In conditional diffusion models, the input to $p_\theta$ also includes a condition to guide the denoising step.

# 4 METHOD

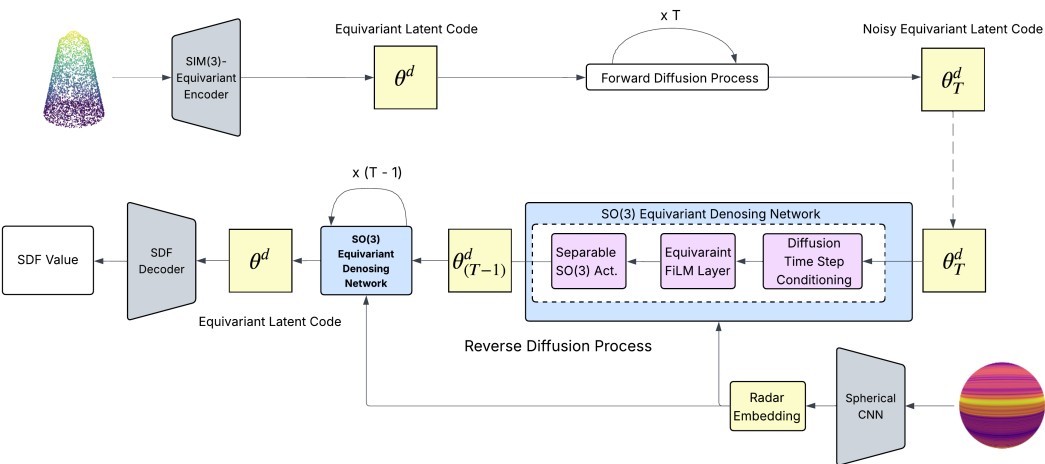

Figure 2: **A schematic representation of our method and its core components.** During training, a SO(3)-Equivariant encoder encodes a point cloud. The encoded latent code is then transformed into a noisy latent code by the forward process. We then denoise the latent code using the reverse diffusion process. For each timestep t, the denoised latent code is predicted by an SO(3) equivariant denoising network, which is conditioned on radar embeddings using equivariant FiLM layers.

Our method consists of training in two stages. First, we train an SO(3)-equivariant encoder-decoder model to generate latent codes for 3D shapes from their point clouds. These latent codes are then used to train a latent diffusion model conditioned on the radar signal of that shape. The complete architecture is described in Figure 2.

## 4.1 SO(3)-EQUIVARIANT SDF MODEL

**Point Cloud Encoder.** We use an SO(3)-equivariant encoder to map point clouds into latent shape encodings. More precisely, given a point cloud $P_{(N \times 3)}$, we generate a latent embedding $\theta = f(P) \in \mathbb{R}^d$, where $d$ is the latent embedding dimension. The encoder $f$ is a Vector Neuron-based encoder (Deng et al., 2021; Lei et al., 2023) that encodes the input point cloud into a global embedding $F$ which is then mapped to four separate components, $(\theta_R, \theta_{inv}, \theta_c, \theta_s)$, where $\theta_R \in \mathbb{R}^{d \times 3}$ is the equivariant latent embedding of the input point cloud, $\theta_{inv} \in \mathbb{R}^{d \times 1}$ is the invariant latent embedding of the input point cloud, and $\theta_c \in \mathbb{R}^{1 \times 3}$ and $\theta_s \in \mathbb{R}^{1 \times 1}$ are the centroid and scale of the input point cloud, respectively. Here, $\theta_R$ and $\theta_c$ are vectors, while $\theta_{inv}$ and $\theta_s$ are scalar components.

**SDF Decoder.** For any given query point, $x \in R^{3 \times 1}$, we canonicalize $x$ by subtracting $\theta_c$ and dividing by $\theta_s$. The channel-wise inner product of the canonicalized $x$ and $\theta_R$ is concatenated with $\theta_{inv}$ and passed through an MLP $\Psi$ to predict the SDF value at $x$.

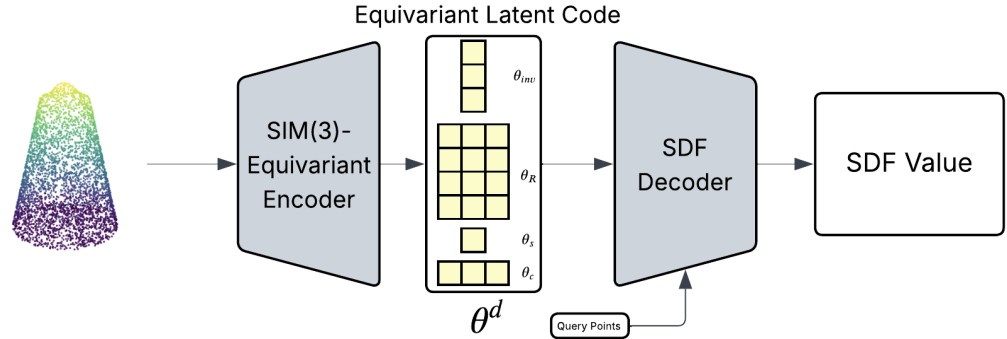

Figure 3: **SO(3)-equivariant SDF Model.** Input point clouds are encoded to an equivariant latent code. SDF decoder predicts sdf value for any query points using the equivariant latent code.

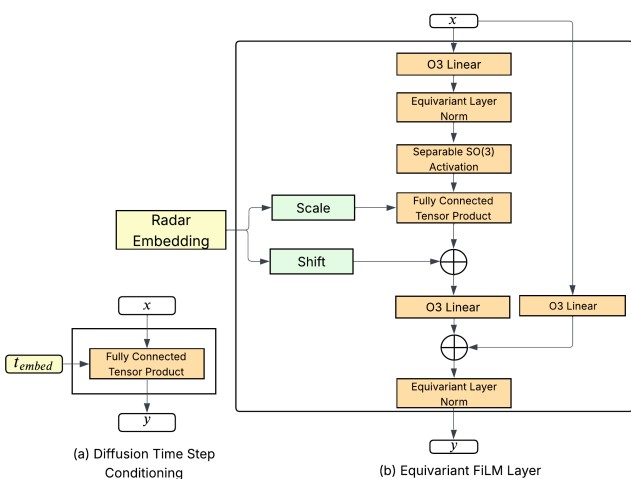

Figure 4: **(a) Time step embedding layer** time step embedding is encoded using a fully connected tensor product layer. **(b) Equivariant FiLM layer**. For conditioning diffusion models on equivariant features, we introduce an equivariant FiLM layer. We modulate the features using a fully connected tensor product and channel-wise addition to keep the hidden features equivariant.

$$\text{SDF}(x) = \Psi([\theta_{inv}, \langle \theta_R, \frac{x - \theta_c}{\theta_s} \rangle]).$$

## 4.2 SO(3)-EQUIVARIANT LATENT DIFFUSION MODEL

We use a denoising diffusion probabilistic model to generate latent embeddings, which are then used to generate 3D shapes by sampling SDF values. Unlike traditional diffusion models, which usually use a U-Net as the denoising network, we designed an SO(3)-equivariant MLP for denoising the latent embeddings. We also use an SO(3)-equivariant spherical CNN to generate the radar signal embeddings. Then, using our novel equivariant FiLM layers, we condition our denoising network on the radar embeddings.

**SO(3) Equivariant Denoising Network.** The denoising network is created using the e3nn library (Geiger & Smidt, 2022). The latent embedding we are trying to denoise consists of scalars and vectors that are equivalent to the irreducible representations of degrees $l = 0$ (scalars) and $l = 1$ (vectors). First, the diffusion time step is encoded using a sinusoidal positional embedding and passed through an MLP to produce a time embedding, $t_{embed}$. The $t_{embed}$ is a scalar feature that is fused with all the intermediate feature representations using a fully connected tensor product as shown in 4 (a). To

condition our model on equivariant radar embedding, we created equivariant FiLM layers, which are described in the next section. The input is passed through the SO(3) equivariant linear, layer norm, and activation layers before and after the FiLM layer. For activation functions, we used separable SO(3) activation functions introduced in the Equiformer architecture(Liao et al., 2023).

**Equivariant FiLM Layer.** Standard FiLM layers first introduced in (Perez et al., 2018) use element-wise product and summation to modulate the scale and shift conditional features. Such layers are used in label-conditioned diffusion models such as (Kumar et al., 2024). Using the standard FiLM layer directly will break the end-to-end equivariance properties of our model. To address this, we design a novel equivariant FiLM layer that allows us to interface between the vector neuron representation and e3nn through the use of Irreps.

As shown in Figure 4 (b), our equivariant FiLM layer first transforms the input using three layers: an equivariant linear layer, a normalization layer, and an activation layer. The hidden features are then conditioned by scale and shift features. We get scale and shift by dividing the conditional feature across the channel dimension. In our case, given the radar embedding $R_L \in \mathbb{R}^{B \times C \times (L+1)^2}$, we get scale and shift $\in \mathbb{R}^{B \times \frac{C}{2} \times (L+1)^2}$. To maintain equivariance, we use a fully connected tensor product (Geiger et al., 2022) for scale and add shift operations across the Irreps dimensions (Cohen et al., 2018). We follow this with another equivariant linear layer and a normalization layer. We also add a skip connection (He et al., 2016). Given a hidden feature h, scale vector $\mu$, and shift vector $\sigma$, we can express the Equivaraint FiLM layer as:

$$\text{EqFiLM}(h|\mu, \sigma) = (h \bigotimes \mu) \oplus \sigma.$$

**SO(3) Equivariant Radar Embedding Network** The radar embedding network takes in a spherical radar signal of resolution ($\omega$, $\phi$), which is expanded into a set of irreducible representations (Irreps) of SO(3) with maximum frequency of $L$ using a truncated Fourier transformation $\mathbb{R}^{C \times \omega \times \phi} \rightarrow \mathbb{R}^{C \times (L+1)^2}$. We then perform SO(3) equivariant group convolutions Cohen et al. (2018) and separable SO(3) activation functions Liao et al. (2023) in multiple stages to progressively refine the representation. See Cohen et al. (2018) for more details about spherical CNNs. Kohler et al. (2025) demonstrates the benefit of using higher frequency spherical harmonics when working with radar signals. Similar to this work, we gradually increase the maximum spherical harmonic degree and the channel dimensionality until they reach $l_{\text{radar}}$ and $d_{radar}$, respectively.

## 5 EXPERIMENTS

### 5.1 FRUSTA DATASET

Due to the lack of large real-world training data for the high-frequency radar setting, we use a physical optics approximation method Balanis (2012) to simulate radar responses for a variety of mesh objects. Specifically, we used the Frusta mesh dataset introduced in Kohler et al. (2023); Sortur et al. (2025). The Frusta dataset consists of roll-symmetric 3D meshes constructed by stacking up to five frusta components of different radii and length, as well as flat plates or hemispheric components on either side of the object. Since the meshes are symmetric, we generate the radar response across the non-symmetric axis over 360 orientations, $\theta \in [0, 2\pi]$. The dataset contains 25k shapes in total. We use 80% of the dataset for training, 10% for validation, and 10% for testing.

### 5.2 BASELINES

We compare our model with two competitive baselines selected to highlight the role of equivariance along the various architecture components, as well as the role of the radar encoding model.

**Diffusion-SDF.** The first baseline is the Diffusion-SDF Chou et al. (2023). Similar to our approach, this model consists of two stages: first, a VAE encodes point clouds into latent representations, then a diffusion model operates on these latent codes for shape generation. The architecture of their diffusion model is the same as the one used in DALLE-2 Ramesh et al. (2022). The original work conditions the diffusion model on partial point clouds through a PointNet encoder. In our adaptation,

we replace the PointNet encoder with a ResNet He et al. (2016) architecture to accommodate radar signal conditioning instead of point cloud inputs.

**Non-equivariant Diffusion.** The second baseline uses the same SO(3)-equivariant SDF model as our method, but the denoising diffusion model is a standard non-equivariant MLP instead of an SO(3) equivariant denoising network, and we use cross-attention layers for conditioning the model on radar embeddings. The MLP consists of four fully connected layers with hidden dimensions $[2056, 4112, 8824, 2056]$. The radar embeddings are generated using a transformer encoder. This baseline allows us to decouple contributions of equivariant modeling in stages 1 and 2 of model training.

### 5.3 EQUIVARIANT DIFFUSION ARCHITECTURE

We set our latent embedding dimension $d = 256$. The SO(3) equivariant denoising network takes an input with 257 channels, which is first projected to 256 channels while expanding the Irreps maximum spherical harmonic degree from $L_{\max} = 1$ to $L_{\max} = 5$ using the equivariant linear and activation layers. The network then consists of three sequential blocks of time steps: conditioning layer, equivariant FiLM layers, and SO(3) activation layer, each with hidden dimensionality of 256 and maximum spherical harmonic degree $L_{max} = 5$. Finally, an equivariant linear and activation layer maps the representation back to match the original input channel dimension and $L_{max}$.

The SO(3) equivariant radar embedding network uses a 3 layer spherical CNN with hidden dimensions $[64, 64, 128]$ and $L_{\max}$' s $[3, 5, 5]$, giving us $d_{\text{radar}} = 128$ and $l_{\text{radar}} = 5$. We set the time embedding to $t_{embed} = 128$.

For our SDF model, we used the SO(3)-equivariant SDF model used by Lei et al. (2023) for their shape prior model.

### 5.4 MODEL TRAINING DETAILS

We train the SO(3)-Equivariant SDF model for 5000 epochs with a batch size of 16 and a learning rate of $1 \times 10^{-4}$ with a decay of 0.3 at 1000, 2000, and 3500 epochs. After training, we select the best validation model to generate latent code for training the diffusion model. The SO(3) equivariant diffusion model was trained for 3137 epochs with a batch size of 8 and a learning rate of $6 \times 10^{-5}$ with a learning rate decay of 0.985 at every 400 training steps.

Table 1: Performance results across various metrics on the Frusta dataset. For each radar signal, we sample 10 point clouds using the diffusion model. The metrics are averaged across all the test samples, $\pm$ indicates std. Error. **Bold** indicates the best performance across all the models for that metric.

| Observability | Model | MMD ($\downarrow$) | TMD ($\uparrow$) | F-Score(1%) ($\uparrow$) |
|---|---|---|---|---|
| **Full** | Equiv. Diffusion | $0.0060 \pm 1e{-}5$ | $\mathbf{0.063} \pm 1e{-}3$ | $\mathbf{0.4279} \pm 1e{-}3$ |
| | Non-equiv. Diffusion | $\mathbf{0.0043} \pm 1e{-}5$ | $0.040 \pm 1e{-}3$ | $0.4027 \pm 1e{-}3$ |
| | Diffusion-SDF | $0.0047 \pm 1e{-}5$ | $0.053 \pm 1e{-}3$ | $0.4245 \pm 1e{-}3$ |
| **Partial** | Equiv. Diffusion | $0.0185 \pm 1e{-}4$ | $\mathbf{0.137} \pm 3e{-}3$ | $0.1549 \pm 6e{-}4$ |
| | Non-equiv. Diffusion | $0.0068 \pm 1e{-}5$ | $0.039 \pm 7e{-}4$ | $0.2455 \pm 1e{-}3$ |
| | Diffusion-SDF | $\mathbf{0.0063} \pm 1e{-}5$ | $0.068 \pm 1e{-}3$ | $\mathbf{0.2807} \pm 1e{-}3$ |

### 5.5 METRICS

To evaluate our model, we consider the following standard metrics from the 3D point cloud reconstruction literature Arshad & Beksi (2023); Chou et al. (2023); Wu et al. (2020).

**Chamfer Distance** Fan et al. (2016) This metric measures the similarity between two point clouds by summing the distances from each point in one set to its nearest neighbor in the other set, in both directions. We use the chamfer distance to compute MMD and TMD metrics described below.

**Minimum Matching Distance (MMD)**    Yang et al. (2019) Achlioptas et al. (2018) MMD measures the reconstruction quality of the generative model by averaging Chamfer distances between the ground truth point cloud and the closest point cloud from the generative model.

**Total Mutual Difference (TMD)**    Chou et al. (2023) This measures the diversity of the generated samples. From k generated samples, we measure the average chamfer distance of each sample to other k-1 samples. TMD is the sum of all the average chamfer distances. Higher TMD values signify that the diffusion model has more diverse output samples. This metric is more significant for the partial observability setting, where multiple possible output shapes are possible.

**F-Score (1%)**    Wang et al. (2018) F-score is computed by taking the harmonic mean of precision and recall. Precision measures the percentage of predicted points that are close to ground truth points, while recall measures the percentage of ground truth points that are close to the predicted points. We use F-Score(1%), where points from one point cloud are considered close to points from another point cloud if they are within 1% distance threshold.

## 5.6   SHAPE RECONSTRUCTION

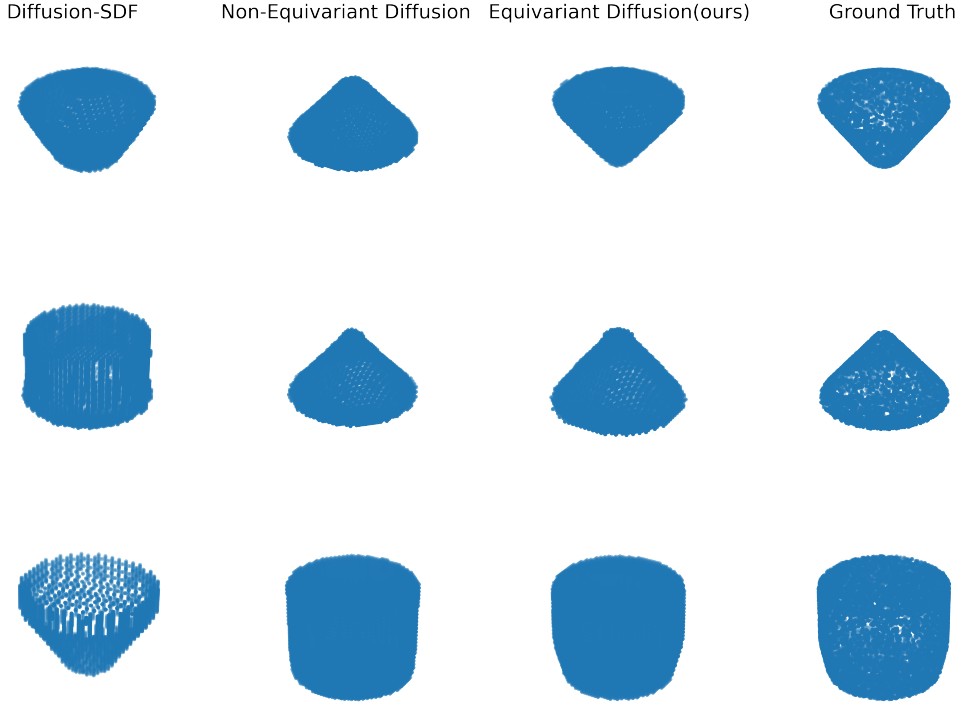

Figure 5: **Generated Frusta Samples.** We generate point cloud samples using the radar signals from the Frusta dataset. We compare the ground truth point clouds with point clouds generated from the Diffusion-SDF, the non-equivariant diffusion model, and our equivariant diffusion model.

Radar signals from the test dataset are used to evaluate the model's performance. For each radar signal, we condition the diffusion model to generate 10 latent codes. We use the latent codes to generate a point cloud associated with that radar signal by sampling query points in a cube and using a threshold of 0.01 to classify a query point as an on-surface point or not. The group of points on the surface generates the final point cloud.

Our model outperforms all the baselines in F-Score(1%) and TMD when we use fully observed radar signals to reconstruct the shape. MMD only measures the reconstruction quality of our output by measuring the chamfer distance to the closest point cloud, while F-score (1%) and TMD take

into account all the sampled point clouds. Figure 5 shows some of the samples generated from the diffusion model. It should also be noted that Diffusion-SDF was trained for around 20,000 epochs, while both Equivariant Diffusion and Non-equivariant diffusion were trained for 3137 and 3799 epochs, respectively. This shows that the equivariant model can give us comparable or better results in far fewer epochs. Even though non-equivariant diffusion does not use an equivariant denoising network, it still uses an equivariant model for shape reconstruction.

### 5.7 SHAPE RECONSTRUCTION FROM PARTIAL OBSERVABILITY

We also test our model with partially observed radar signals. This is done by masking 10-40% of the radar signals before generating the latent code from the diffusion model. Due to the masking of the radar signals, the complete output shape cannot be directly inferred from the radar signal, leaving uncertainty in the output shape. Shapes generated from partial radar signals should also reflect this uncertainty, and we try to measure that by comparing the TMD metric described above. The performance results of our method and the baselines are summarized in 1.

While all of the models drop performance across all the metrics, we can see that our method still has the highest TMD despite having higher MMD and F-Score (1%). This shows us that while the baseline methods conditioned on partially observed radar signals are sampling similar point clouds to the ground truth point cloud, our method is sampling a more diverse range of point clouds and is not overfitting to just one single point cloud, as it should be the case when generating complete shapes from partially observed radar signals.

## 6 CONCLUSION

In this work, we present an equivariant diffusion model for the inverse problem of reconstructing the 3D geometry given its radar signature. We also introduced an equivariant FiLM layer for conditioning diffusion models on radar signals and developed a new architecture for conditional latent diffusion models that are equivariant to SO(3) group transformations, done on the condition which is the radar signal is in this work. Our method outperforms strong baselines in F-score (1%) and TMD metrics and has competitive results in the MMD metrics. We also show that when provided with a partially observed radar signal, our model gives more diverse point clouds compared to the baselines. We also show that our equivariant model requires fewer training epochs compared to non-equivariant methods.

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
