# OpenReview forum: "Equivariant Diffusion for The Inverse Radar Problem"
_ICLR.cc/2026/Conference — Submitted to ICLR 2026_

### Official Review · Reviewer_sAnG · 2025-10-25

**Soundness:** 3
**Presentation:** 2
**Contribution:** 3
**Rating:** 6
**Confidence:** 3

**Summary:**

The paper presents a 3D reconstruction method from radar signals, which is based on a latent diffusion model composed of an SO(3)-equivariant autoencoder and an SO(3)-equivariant denoiser model. Experiments on synthetic (Frusta) data show that the proposed model can obtain good shape reconstruction, outperforming baselines that do not use both an equivariant autoencoder and denoiser. The paper introduces a conditioning network that can incorporate information about the radar signal into the diffusion denoising network in an equivariant manner, leveraging ideas from spherical CNNs.

**Strengths:**

- Presents a new diffusion approach for solving a challenging radar inverse problem involving spherical measurements and 3D point cloud signals.
- Designs a model where every component is equivariant: autoencoder, diffusion denoising model, and measurement conditioning module.
- Experiments on synthetic data show that the model can recover roll-symmetric 3D shapes from the measurements.

**Weaknesses:**

- The presentation of the paper should be improved
    - the radar forward model is not explained, making the paper not very accessible to a general ML audience. I would expect the paper to include a more self-contained mathematical description of the forward problem, at least in an appendix.
    - mathematical notation is not consistent:
           - subindices are sometimes written with italics (eg $d_{\mathrm{radar}}$, sometimes not (eg $L_{max}$)
           - Mathbb is lacking on some spaces $x\in R^3$
           - the use of uppercase/lowercase doesn't seem to follow any pattern. I would expect vectors in lowercase and matrices in uppercase.
           - the $\mathrm{SDF}$ equation has double parenthesis, and the inner product is not well defined for matrices.
    - citation style is used inconsistently: \citep and \cite are not used correctly in many places in the paper.
           - I believe the presentation of the method could be improved: at the moment, the presentation is divided across sections 3, 4 and 5, and feels a bit repetitive, where none of the sections goes into enough detail.

- Only synthetic experiments are performed, and it is unclear if the method can work well in realistic settings: the network is trained and evaluated on similar simplistic 3D point clouds, and the measurement process is synthetic. It is not clear from the paper whether measurement noise is considered.

- Another figure should be included illustrating the sampling diversity in the case of partial measurements, which is the main motivation of the paper for using diffusion models.

**Questions:**

- Why do you use different equivariant architectures/paradigms for the autoencoder and the denoising model? Why not using the same paradigm?
- Why are radar measurements defined on the sphere? This should be better explained to a general ML audience in the paper.

---

> ### Author Response · Authors · 2025-12-03
>
> Thank you for your review. We have address your comments and questions here as well
>
>
> * *the radar forward model is not explained, making the paper not very accessible to a general ML audience. I would expect the paper to include a more self-contained mathematical description of the forward problem, at least in an appendix.*
>
>
> We agree that the radar forward model deserves a more thorough treatment for readers unfamiliar with radar signal processing. In the final version, we will add a detailed, self-contained mathematical description of the radar forward model in the appendix. Thank you for pointing this out.
>
>
> * *Why do you use different equivariant architectures/paradigms for the autoencoder and the denoising model? Why not using the same paradigm?
> We chose the Vector Neuron architecture for generating equivariant latent codes for 3D shapes because it has been shown to produce high-quality reconstructions with excellent equivariance properties [1][2][3]. Using Vector Neurons in the autoencoder ensures that the learned latent space is well-structured and preserves geometric equivariance.*
>
> We chose an e3nn-based architecture because we needed to incorporate Spherical CNN feature embeddings to condition the diffusion model on radar signals. A spherical CNN-based architecture for radar modelling has shown state-of-the-art results for the forward radar modelling task in prior work[4] compared to other architectures. The e3nn framework integrates seamlessly with Spherical CNNs while maintaining SO(3) equivariance throughout the denoising process.
>
> In summary, we chose Vector Neurons for high-quality latent representation learning, and e3nn + Spherical CNNs for effective radar-conditioned diffusion. This combination is motivated by the strengths of each approach in their respective domains.
>
> [1] Jiahui Lei, Congyue Deng, Karl Schmeckpeper, Leonidas Guibas, and Kostas Daniilidis." Efem: Equivariant neural field expectation maximization for 3d object segmentation without scene supervision", IEEE / CVF Computer Vision and Pattern Recognition Conference (CVPR), 2023.
>
> [2] Jingyun Yang, Congyue Deng, Jimmy Wu, Rika Antonova, Leonidas Guibas, and Jeannette Bohg. "Equivact: Sim(3)-equivariant visuomotor policies beyond rigid object manipulation", IEEE International Conference on Robotics and Automation (ICRA), 2024.
>
> [3] Yang, Jingyun, Zi-Ang Cao, Congyue Deng, Rika Antonova, Shuran Song, and Jeannette Bohg. "EquiBot: SIM(3)-Equivariant Diffusion Policy for Generalizable and Data Efficient Learning", Conference on Robot Learning (CoRL), 2024
>
> [4] Colin Kohler, Purvik Patel, Nathan Vaska, Justin Goodwin, Matthew C. Jones, Robert Platt, Rajmonda S. Caceres, and Robin Walters. Bridging equivariant GNNs and spherical CNNs for structured physical domains. In The Thirty-ninth Annual Conference on Neural Information Processing Systems, 2025.
>
> * *Why are radar measurements defined on the sphere? This should be better explained to a general ML audience in the paper.*
>
> We appreciate this question, as it highlights an important point that deserves clarification. To address the potential confusion: we do not explicitly define radar measurements on the sphere. the spherical structure is an inherent characteristic of the radar function. The radar function depends on the sensor viewing angle and therefore it is a spherical function.
> Specifically, our radar simulator uses viewing directions parameterized by spherical coordinates,  α ∈ [0, π] and ϕ ∈ [0, 2π] to measure radar responses from different viewpoints around the 3D geometry. These viewing directions inherently form a sphere, which is why the resulting radar measurements are naturally defined on a spherical domain.
>
> We understand that this connection may not be immediately clear to readers unfamiliar with radar signal processing. In the revised paper, we will clarify this relationship more explicitly.

---

### Official Review · Reviewer_nqi6 · 2025-10-27

**Soundness:** 3
**Presentation:** 3
**Contribution:** 3
**Rating:** 4
**Confidence:** 3

**Summary:**

This paper investigates the reconstruction of 3D objects, particularly focussing on partial radar signals. The authors propose a two-stage, SO(3)-equivariant pipeline. Stage 1 involves training an SO(3) equivariant encoder–decoder model which is used to generate latent encodings from point clouds. Stage 2 trains an SO(3)-equivariant denoise operating on the latent encodings and is conditioned on spherical radar embeddings via a novel equivariant FiLM layer; conditioning features come from a spherical CNN. The full generative pipeline is end-to-end SO(3)-equivariant. The authors run a series of experiments on a simulated Frusta dataset due to a lack of high-frequency real-world data. The model is compared to two baselines, which do not have end-to-end equivariance constraints using a minimum matching distance (MMD) and F1 score for reconstruction accuracy, and total mutual difference (to show diversity/uncertainty benefits). The authors model beats the two baselines in terms of F1 score for the fully observed radar signals, however performs slightly worse than both baselines (for F1 score) for partially observed radar signals. Their equivariant model does have larger sample diversity as shown by the TMD scores, implying it is more uncertain about the true shape. Overall the method is interesting, performs at least as good at the baselines in the fully observed case, and produces richer sample diversity in all cases.

**Strengths:**

The end-to-end equivariance pipeline is well motivated for the inverse radar mapping problem, and incorporating this into both the encoder/decoder model, and the diffusion model is a good idea. The equivariant FiLM layer is also a nice addition to allow the full pipeline to be equivariant to SO(3) rotations.

One would expect in under-sampled regimes (partial observations) that there should be a larger amount of uncertainty about the true shape, hence the diversity of sampled from the diffusion model would be higher. This is indeed the case as shown in table 1.

The improved efficiency over the Diffusion-SDF model is substantial, although the non-equivariant baseline is roughly as efficient as this proposed model too.

In figure 5, the generated sampled of the equivariant diffusion model appear much closer to the ground truths than the baselines.

**Weaknesses:**

Firstly, from table 1 the equivariant diffusion samples appear to not perform any better than the baselines on average. Specifically while for the full observations the F1 performance in the best, this is only be a very minor amount, and the MMD scores of the equivariant model are the worst of all models for both the partial and full observed datasets.

Second, in the abstract and introduction the authors mention the difficulty with partially observed data, and dealing with the uncertainty that arises from this. However their results in the partial test case are significantly worse for reconstruction accuracy (MMD and F1) than both baselines. The sample diversity in higher for the authors model which I agree one would think is due to the uncertainty of the true shape, but the significantly lower (0.1549 vs 0.2807) F1 accuracy is not insignificant.

Third, the dataset is synthetic which is understandable and not necessarily a bad thing, however this specific dataset used contains only axially symmetric data. This seems like it would potentially help the performance of SO(3)-equivariant models, while true data may not be symmetric at all. Testing on non-symmetric data (even if still synthetic) would be a benefit here.

Minor issues, figure 5 is not fully explained, is this full or partially observed data? Some in text citations are inconsistently formatted such as this sentence "(Esteves et al., 2018)(Cohen et al., 2018) uses it for 3D
shape classification." Other sections use \citet style.

**Questions:**

- Is the increase sample diversity hurting the reconstruction accuracy? I.e is there a way to test if the model is perhaps being under-confident in the shape predictions?

- Do you have tests of your model on non-symmetric datasets compared to the baseline models?

- Do you have any reasons or results as to why you model seems to perform worse on the MMD baselines even for the full observation testing?

- Is there results on any real-world datasets?

---

> ### Author Response · Authors · 2025-12-03
>
> * *Is the increase sample diversity hurting the reconstruction accuracy? I.e is there a way to test if the model is perhaps being under-confident in the shape predictions?*
>
> When our model is tested with partially observed radar signals, MMD increases, F-score (1%) decreases relative to the fully observed setting, and TMD also increases. This indicates higher sample diversity, but it does not necessarily imply a reduction in reconstruction quality, because a partially observed signal can correspond to multiple plausible shapes, while mmd and F-score(1%) compare each reconstruction against a single ground-truth shape.
>
> * *Do you have tests of your model on non-symmetric datasets compared to the baseline models?*
>
> * *Is there results on any real-world datasets?*
>
> To address this concerns we also ran our model on a non-symmetric dataset consisting of airplane point clouds from Manifold-40 Dataset. Due to the limited time of rebuttal period we were only able to train one single baseline to compare against. We obtained similar results to those presented in the Frusta dataset paper. In fully observed radar signals our model outperforms Diffusion-SDF in F-Score(1%) and TMD metrics, and in partial radar signals, our model has the highest TMD compared to the baseline due to the same reason we outlined in the paper. The higher TMD scores show our model’s capabilities in capturing the uncertainty in the reconstruction process when the input radar signal is only partially observed.
>
> ### Aiplanes Dataset With Fully Observed Radar Signal
>
> | Model | MMD ($\downarrow$) | TMD ($\uparrow$) | Fscore(1%)  ($\uparrow$) |
> |--------|--------|-------|-------|
> | Equiv. Diffusion |  0.0068  ± 3e-4 | **0.0348 ± 6e-2** | **0.6434 ± 8e-3** |
> | Diffusion-SDF.   | **0.0019 ± 5e-5** | 0.0230 ± 7e-4     | 0.4517 ± 6e-3 |
>
> ### Aiplanes Dataset With Partially Observed Radar Signal
>
> | Model | MMD ($\downarrow$) | TMD ($\uparrow$) | Fscore(1%)  ($\uparrow$) |
> |--------|--------|-------|-------|
> | Equiv. Diffusion | 0.0190 ± 4e-4 | **0.0129 ± 6e-3** | 0.1960 ± 3e-3 |
> | Diffusion-SDF   | **0.0019 ± 5e-5** | 0.0189 ± 1e-3. | **0.4395 ± 4e-3** |

---

### Official Review · Reviewer_KVpF · 2025-10-30

**Soundness:** 2
**Presentation:** 2
**Contribution:** 1
**Rating:** 2
**Confidence:** 4

**Summary:**

This paper introduces a radar-conditioned SO(3)-equivariant latent diffusion model for reconstructing 3D volumes from radar signals. Following the latent diffusion paradigm, the proposed approach is trained in two stages. First, an SO(3)-equivariant encoder–decoder is trained using vector neuron architectures to learn a structured latent space. Then, an SO(3)-equivariant diffusion model is trained to operate within this latent space. To enable rotationally equivariant conditioning, the authors propose an equivariant FiLM layer that modulates the diffusion process directly in the Irreps space. Experimental results on a simulated dataset demonstrate that the proposed method sometimes outperforms existing baselines.

**Strengths:**

The paper is well-written and easy to follow. The main idea and contributions are explained well.

**Weaknesses:**

The paper has several limitations that make it unsuitable for acceptance at this conference.

1. The contribution of the paper seems incremental. While SO(3)-equivariant neural networks are well-established in the literature, this work primarily applies them within a diffusion model framework without introducing a fundamentally new concept or significant methodological advance.

2. The experiments are limited to a simplified simulated dataset, which does not convincingly demonstrate the generalization ability or practical value of the proposed approach. Moreover, the baselines are inadequate; the authors should compare against a broader range of methods, including both standard deep learning approaches and traditional radar imaging methods. Evaluating the method on more challenging and realistic datasets for shape reconstruction would improve the paper.

**Questions:**

In the experimental section, the authors state: “Due to the lack of large real-world training data for the high-frequency radar setting, we use a physical optics approximation method to simulate radar responses for a variety of mesh objects.”
This raises a fundamental concern regarding the practical relevance of the proposed approach. If no real-world training dataset is available, it is unclear how this supervised method could be applied in real scenarios. Without any experiments on real radar data, it is hard to assess the real-world performance, generalization, or usefulness of the proposed model.

---

> ### Author Response · Authors · 2025-12-03
>
> Thank you for your review. We have addressed your comments and questions here
>
> * *The contribution of the paper seems incremental. While SO(3)-equivariant neural networks are well-established in the literature, this work primarily applies them within a diffusion model framework without introducing a fundamentally new concept or significant methodological advance.*
>
> We introduce Equivariant FiLM layers, a novel architectural component for conditioning equivariant representations in a way that preserves SO(3) symmetry. Our work also presents a way to combine the vector neuron framework with  spherical CNNs, which has not been studied before in the literature
>
>
> * *In the experimental section, the authors state: “Due to the lack of large real-world training data for the high-frequency radar setting, we use a physical optics approximation method to simulate radar responses for a variety of mesh objects.” This raises a fundamental concern regarding the practical relevance of the proposed approach. If no real-world training dataset is available, it is unclear how this supervised method could be applied in real scenarios. Without any experiments on real radar data, it is hard to assess the real-world performance, generalization, or usefulness of the proposed model.*
>
> Although our dataset is generated using a radar simulator, the radar simulations we used are high-quality and high-resolution simulations  and have been widely used for far-field radar modeling.[1], [2], [3]
>
>
> [1] - InvRT: Solving Radar Inverse Problems with Transformers, R Muthukrishnan, J Goodwin, A Kern, N Vaska, Rajmonda S Caceres, AAAI, 2023.
>
> [2] - Reducing the Sensitivity of Neural Physics Simulators to Mesh Topology via Pretraining, N. Vaska, J. Goodwin, R. Walters, R. Caceres, ICASSP 2025.
>
> [3] - Do you see the shape? Diffusion Models for Noisy Radar Scattering Problems, N. Sortur, J. Goodwin, R. Caceres, R. Walters, Frontiers in Probabilistic Inference Workshop at ICLR 2025.

---

### Official Review · Reviewer_hS9M · 2025-11-02

**Soundness:** 3
**Presentation:** 3
**Contribution:** 3
**Rating:** 4
**Confidence:** 4

**Summary:**

This paper proposes a radar-conditioned SO(3)-equivariant latent diffusion model for reconstructing 3-dimensional shapes from radar signals. The aim is to use group equivariance to reduce uncertainty in the solution arising from incomplete observations and inherent radar noise (irreducible uncertainty), thereby generating more plausible, diverse and consistent three-dimensional shapes (represented implicitly via SDFs). The proposed method employs a two-stage training strategy: First, SO(3)-equivariant latent codes are learnt for the 3D shape. Then, an isometry-preserving denoising diffusion model is trained on these latent representations. Specifically, a SDF serves as the output representation, with physical optics (PO) simulations modelling high-frequency radar responses and range profiles (spherically parameterised) used for conditioning. The denoising network employs an SO(3)-equivariant MLP implemented via e3nn, while the radar encoder uses a spherical CNN. Finally, the FiLM layer is modified to preserve equivariance in the irreps space.

**Strengths:**

1. The idea of leveraging symmetry biases explicitly in reconstruction and using diffusion models for uncertainty estimation is interesting. Ideally, embedding SO(3) equivariance as a prior throughout the entire generation pipeline (i.e. the encoder, denoising network and conditioner) would reduce sample complexity, enhance data efficiency and ensure rotational consistency (where rotation constitutes an inherent symmetry of the problem).

2. The proposed 'equivariant FiLM layer' performing conditional modulation in irreps space is an interesting solution that combines common conditioning techniques (FiLM) with an approach based on representations. For high-frequency radar, the integration of SDF representations with PO models has led to the construction of a relatively comprehensive simulation dataset (Frusta dataset) and evaluation workflow.

**Weaknesses:**

1. The mathematical proof or sufficient justification for the equivariance of FiLM is inadequate. For example, line 249: ‘using fully connected tensor product and channel-wise addition to preserve equivariance’, but lacks a formal proof or explanation of the conditions under which FiLM (scaling + bias) still satisfies strict SO(3) on Irreps (e.g., which terms must be scalars, how vectors are handled, etc.).

2. The equivariance argument for embedding time step t_embed remains unclear: embedding the time step as a scalar via tensor product requires explicit clarification of how this operation interacts with different-order representations ($l=0,1$) while preserving isometry.

3. The paper brings together three well-studied ideas: equivariant neural networks for three-dimensional representation, latent diffusion modelling and spherical convolutional neural networks (CNNs). There has been limited innovation in either the theoretical or engineering aspects.

**Questions:**

Ablation experiments are necessary, for example, the independent contributions of the equivariant FiLM, equivariant denoiser, and equivariant encoder were not demonstrated.

How can FiLM be built for other symmetry groups, such as permutation or a mixture of rotation and shifts? Does the design of FiLM's symmetry group strictly depend on (or should match) the symmetry of the radar data?

---

> ### Author Response · Authors · 2025-12-03
>
> Thank you for your review. We have address your comments and questions here as well
>
>
> * *The mathematical proof or sufficient justification for the equivariance of FiLM is inadequate. *
>
> 	Here is the Proof for the Equivariant FiLM layer.
> 		EqFiLM(h|μ, σ) = (h ⊗ μ) ⊕ σ
> Here h, μ, σ are features in SO(3) irreps space,
> Let R ∈SO(3) and D(R) denote the corresponding wigner D-matrix acting on the Layer
>
> 		EqFiLM(D(R)h|D(R)μ, D(R)σ) = ((D(R)h) ⊗ (D(R)μ)) ⊕ (D(R)σ)       (1)
>
> Using the equivariance property of the fully connected tensor product given here [1] in equation (1), we get
>
> 		EqFiLM(D(R)h|D(R)μ, D(R)σ) =(D(R)(h ⊗ μ)) ⊕ (D(R)σ)		(2)
>
> 		EqFiLM(D(R)h|D(R)μ, D(R)σ) =D(R) ((h ⊗ μ) ⊕ σ)               (3)
>
> 		EqFiLM(D(R)h|D(R)μ, D(R)σ) =(D(R)*EqFiLM(h|μ, σ)             (4)
> Equation (4) proves that our EqFiLM layers are equivariant under  R ∈SO(3)
>
>
> * *The equivariance argument for embedding time step t_embed remains unclear: embedding the time step as a scalar via tensor product requires explicit clarification of how this operation interacts with different-order representations () while preserving isometry.*
>
> To clarify our design, we embed the time step t as a scalar, which is then fused with the input feature x through a fully connected tensor product layer. This tensor product operation is equivariant to SO(3) transformations and preserves isometry throughout the network. Also, the fully connected tensor product uses Clebsch-Gordan (CG) coefficients to couple the scalar t_embed with different-order representations. Our choice of using a fully connected tensor product layer ensures that the time step information is fused with all orders of representation in a way that respects the underlying SO(3) symmetry.
>
> * *The paper brings together three well-studied ideas: equivariant neural networks for three-dimensional representation, latent diffusion modelling and spherical convolutional neural networks (CNNs). There has been limited innovation in either the theoretical or engineering aspects.*
>
> While Equivariant Neural Networks, Latent diffusion models and Spherical CNNs are well studied ideas, To our knowledge, there has been limited work [2] on combining equivariant diffusion models with SDF representations for 3D shape generation. Our method uses an equivariant SDF representation within a latent diffusion framework for 3D shape generation from radar signals, which to our knowledge has not been explored. Our paper also introduces equivariant FiLM Layer for conditioning diffusion model on radar signal while preserving SO(3) symmetry during denoising steps.
>
>
> * *Ablation experiments are necessary, for example, the independent contributions of the equivariant FiLM, equivariant denoiser, and equivariant encoder were not demonstrated.*
>
> Ablating the equivariant FiLM or equivariant encoder by replacing them with non-equivariant counterparts is theoretically equivalent to breaking equivariance entirely, since a single non-equivariant layer in the pipeline breaks the end-to-end equivariance. Such an ablation would not isolate the contribution of these specific components since we won’t know if the  performance difference is due to  the component itself or due to the loss of equivariance.
> However, our paper does include an ablation for the equivariant denoiser: the non-equivariant diffusion  baseline present in the paper directly replaces the equivariant denoiser with a non-equivariant variant, isolating the contribution of equivariance in the denoising process while keeping the rest of the pipeline same.
>
> * *How can FiLM be built for other symmetry groups, such as permutation or a mixture of rotation and shifts? Does the design of FiLM's symmetry group strictly depend on (or should match) the symmetry of the radar data?*
>
>
> We have not extended FiLM to other symmetry groups because these symmetries are not present in our data and application domain. Developing equivariant FiLM layers for other symmetry groups is an interesting direction, but it is beyond the scope of this work. We view this as future work.
>
> Yes, the design of FiLM's symmetry group must match the symmetry of the radar data. We want the denoising process in the diffusion model to be conditioned on the radar features  while preserving its inherent SO(3) symmetry structure. If we used a FiLM layer with a different symmetry group, we would break the end-to-end equivariance of our architecture. Thus, matching the symmetry groups is not just a design choice, it is a requirement for maintaining end-to-end equivariance during the conditional generation process.
>
> [1] https://docs.e3nn.org/en/stable/api/o3/o3_tp.html
>
> [2] Jiahui Lei, Congyue Deng, Karl Schmeckpeper, Leonidas Guibas, and Kostas Daniilidis." Efem: Equivariant neural field expectation maximization for 3d object segmentation without scene supervision", IEEE / CVF Computer Vision and Pattern Recognition Conference (CVPR), 2023.

---

### Author Response · Authors · 2025-12-03

We appreciate the reviewers’ time and constructive comments, and thank the emergency AC for working under the time constraints to review our work.

Our paper introduces a novel approach to solving the inverse diffusion problem. We used SO(3) equivariant Latent diffusion model for reconstructing 3D shapes from fully and partially observed radar signals. Our model outperforms baselines in key metrics in both fully observed and partially observed radar settings.

**Reviewer hS9M**

Praise: the Equivariant FiLM layer introduced by the paper, and also how we leverage symmetry to solve this problem.

Concern: Proof of equivariance for Equivariant FiLM Layer, Using the tensor product to embed time step, Ablation for our method, Equivariant FiLM only for SO(3) symmetries

 Response: We have provided the proof for the Equivariant FiLM layer. We also justify our architecture design choices for embedding the time step to maintain end-to-end equivariance. We also showed that our non-equivariant Diffusion baseline is an ablation of our method and why we designed the equivariant FiLM layer only for SO(3) symmetries.

**Reviewer KVpF**

Praises: Well written and easy to follow

Concern: Novelty, Lack of real-world data

Response: We pushed back the concern of novelty by pointing the Equivariant FiLM layer introduced by the paper, and also the lack of work on combining vector neuron framework with spherical CNN's. We have also added results on a real-world dataset consisting of airplanes from Manifold-40 dataset and compared it with the baseline below.

**Reviewer nqi6**

Praises: end-to-end equivariance pipeline, our methods' reconstruction quality and efficiency compared to the Diffusion-SDF baseline.

Concerns: Lack of results on real-world non-symmetric datasets.

Response:  We have provided results on a real-world dataset consisting of non-symmetric airplanes from Manifold-40 dataset and compared it with the baseline below, and showed similar improvements as the Frusta dataset provided in the paper.



**Reviewer sAnG**

Praises: New approach for solving inverse radar problem, end-to-end equivariance pipeline, Experiments on Frusta dataset

Concern: Inconsistencies in mathematical notations and citations, lack of a real-world dataset, and questions regarding combining vector neuron architecture with spherical CNN, and also about defining the radar signal on sphere

Response: We will ensure that all the inconsistencies presented in the paper are removed from the final draft and also add a self-contained mathematical description of the radar forward model in the appendix. We also justified our choice of using Vector Neurons for high-quality latent representation learning, and e3nn + Spherical CNNs for effective radar-conditioned diffusion.  We also explained how the spherical structure of radar signals is an inherent characteristic and not a design choice by us.


We would also like to add the results from the experiments conducted on the airplane dataset from Manifold-40, which is a real-world dataset with non-symmetric objects.



### Aiplanes Dataset With Fully Observed Radar Signal

| Model | MMD ($\downarrow$) | TMD ($\uparrow$) | Fscore(1%)  ($\uparrow$) |
|--------|--------|-------|-------|
| Equiv. Diffusion |  0.0068  ± 3e-4 | **0.0348 ± 6e-2** | **0.6434 ± 8e-3** |
| Diffusion-SDF.   | **0.0019 ± 5e-5** | 0.0230 ± 7e-4     | 0.4517 ± 6e-3 |

### Aiplanes Dataset With Partially Observed Radar Signal

| Model | MMD ($\downarrow$) | TMD ($\uparrow$) | Fscore(1%)  ($\uparrow$) |
|--------|--------|-------|-------|
| Equiv. Diffusion | 0.0190 ± 4e-4 | **0.0129 ± 6e-3** | 0.1960 ± 3e-3 |
| Diffusion-SDF   | **0.0019 ± 5e-5** | 0.0189 ± 1e-3. | **0.4395 ± 4e-3** |

This result further strengthens our model's superiority in reconstruction quality in a fully observed setting, and its capabilities of capturing uncertainties in shape reconstruction when provided with a partially observed radar signal.

Due to time constraints, we were unable to finish the training of non-equivariant diffusion baseline, but once we have results for that, we can include the complete results in the final revised draft.


Given the addition of experiments on another real-world dataset and addressing the concerns stated above, we believe our paper fully meets the acceptance criteria for ICLR conference.

---

### Meta-Review · Area_Chair_11yn · 2026-01-04

**Summary:**

The paper received mixed initial reviews, with scores of 6, 4, 4, and 2, and the overall sentiment among reviewers leaned negative. While several reviewers appreciated the end-to-end SO(3)-equivariant design and the technical integration of equivariant diffusion with radar-conditioned shape reconstruction, substantial concerns were raised regarding the novelty and practical relevance of the contribution. In particular, reviewers questioned whether the work goes beyond applying existing equivariant and diffusion techniques, and noted the absence of real-world radar experiments and the limited scope of baseline comparisons.

In the rebuttal, the authors provided clarifications on equivariance guarantees, added experiments on a non-symmetric synthetic dataset, and improved presentation and technical explanations. However, these responses did not fully address the core concerns shared by multiple reviewers. Most notably, the paper still relies entirely on synthetic data, with no validation on real radar measurements, and the baseline coverage remains restricted. As a result, reviewer opinions are unlikely to shift positively, and the AC anticipates that final scores will remain largely unchanged or even decrease, converging toward rejection (see detailed discussion in Reviewer Concerns and Reviewer Scores).

From the AC’s perspective, while the overall idea is reasonable and technically promising, the AC shares the key concerns raised by the reviewers. The lack of real-data evaluation, the narrow set of baselines, and unresolved questions about conceptual novelty make it difficult to assess the practical impact and significance of the work at this stage. Given this convergence of concerns, the AC therefore recommends rejection.

**Reviewer Concerns:**

### Reviewer hS9M (Score: 4)

- The reviewer raised concerns about the rigor of the equivariance claims, specifically requesting a formal proof for the proposed equivariant FiLM layer and clarification of how diffusion time-step embeddings preserve equivariance. They also questioned the overall novelty, viewing the method as a composition of existing components (equivariant networks, diffusion models, spherical CNNs), and requested clearer ablations isolating the contribution of individual equivariant components.
- In the rebuttal, the authors provided a formal equivariance proof for the FiLM layer, clarified that diffusion time-step embeddings are scalar quantities injected through equivariant operations, and argued that comparisons to non-equivariant baselines already serve as meaningful ablations. They also clarified the decision to restrict equivariance to SO(3), given the symmetry properties of the data.

---

### Reviewer KVpF (Score: 2)

- The reviewer expressed strong concerns about limited novelty, characterizing the work as an application of known equivariant techniques within diffusion models. They also questioned the practical relevance due to the exclusive use of synthetic data, the absence of real-world radar experiments, and limited baseline comparisons.
- In response, the authors emphasized the novelty of the proposed equivariant FiLM conditioning and the integration of multiple equivariant paradigms within a diffusion framework. They justified the reliance on high-fidelity radar simulation by citing precedent in the radar literature, but did not add real-world radar experiments or more baselines.

---

### Reviewer nqi6 (Score: 4)

- The reviewer raised concerns about reduced reconstruction accuracy compared to baselines, particularly under partial observation settings, potential bias arising from symmetric synthetic datasets, and the interpretation of diversity–accuracy trade-offs in the reported results.
- In the rebuttal, the authors clarified that lower accuracy reflects inherent ambiguity under partial observability, added experiments on a non-symmetric dataset (Manifold-40 airplanes), and argued that increased output diversity is a desirable property in ambiguous inverse problems.

---

### Reviewer sAnG (Score: 6)

- The reviewer raised multiple concerns related to clarity and presentation, including inconsistent notation, insufficient explanation of the radar forward model, and unclear architectural choices (such as the use of different equivariant paradigms for the encoder and denoiser). They also questioned the absence of real-data validation.
- In the rebuttal, the authors committed to improving presentation, adding a clearer description of the radar forward model, justifying architectural choices, and refining figures and explanations.

**Reviewer Scores:**

### Reviewer hS9M

- **Original score:** 4
- **Predicted final score:** 4–6
- **Rationale:** The rebuttal addresses the reviewer’s technical concerns regarding equivariance proofs and time-step conditioning. However, conceptual concerns about novelty may persist. Overall, there is a possibility of a score increase, but it is not very likely.

---

### Reviewer KVpF

- **Original score:** 2
- **Predicted final score:** 2
- **Rationale:** The reviewer’s core concerns regarding limited novelty and the lack of real-world radar validation are unlikely to be resolved. The novelty concern is largely conceptual, and no real-data experiments were added in the rebuttal, making a score change unlikely.

---

### Reviewer nqi6

- **Original score:** 4
- **Predicted final score:** 4
- **Rationale:** The addition of results on a non-symmetric dataset and clarification of the diversity–accuracy trade-off directly respond to the reviewer’s concerns. However, no real-world data is provided, and the new results exhibit similar quantitative limitations as the original experiments. The reviewer is therefore likely to maintain their score.

---

### Reviewer sAnG

- **Original score:** 6
- **Predicted final score:** 4
- **Rationale:** The reviewer was mildly positive with relatively low confidence. While the rebuttal addresses clarity and presentation issues, it does not resolve the reviewer’s concern about the absence of real-world data. As a result, a score decrease is likely.

---

### Decision · Program_Chairs · 2026-01-26

Reject